

# Auroral Classification Ergonomics and the Implications for Machine Learning

Derek McKay[1] and Andreas Kvammen[2]

[1]NORCE Norwegian Research Centre AS, Tromsø, Norway
[2]Department of Physics and Technology, UiT – The Arctic University of Norway, Tromsø, Norway

**Correspondence:** D. McKay (demc@norceresearch.no)

**Abstract.** The machine learning research community has focused greatly on bias in algorithms and have identified different manifestations of it. Bias in the training samples is recognised as a potential source of prejudice in machine learning. It can be introduced by human experts who define the training sets. As machine learning techniques are being applied to auroral classification, it is important to identify and address potential sources of expert-injected bias. In an ongoing study, 13 947 auroral images were manually classified with significant differences between classifications. This large data set allowed identification of some of these biases, especially those originating as a result of the ergonomics of the classification process. These findings are presented in this paper, to serve as a checklist for improving training data integrity, not just for expert classifications, but also for crowd-sourced, citizen science projects. As the application of machine learning techniques to auroral research is relatively new, it is important that biases are identified and addressed before they become endemic in the corpus of training data.

## 1 Introduction

Each year, the all-sky cameras in the Arctic and Antarctic regions are collecting several millions of images of the sky. These contain a plethora of atmospheric and astronomical phenomena including, of particular interest to the authors, manifestations of the aurorae. Having computer algorithms to pick out interesting features, or features where there is potential for scientifically interesting phenomena, is helpful for their research.

Examination of what can be done using machine learning for such interests has been pursued, and there are other groups doing the same. Ideally, autonomous software would take a set of images and identify those which contain aurorae and, in these cases, which morphological types are present (patchy, arcs, diffuse, discrete, etc.).

Yet, although algorithms for identification of visual features have made remarkable progress, these tend to be "exceptionally data-hungry". It is well-established that it is expensive and tedious to produce large, labelled training datasets, especially in cases where expert knowledge is required (Yu et al., 2015, e.g.).

Although initial attempts have been made to undertake automatic auroral classification, these have not been particularly successful (low prediction rates), or useful (good classification, but the categories are so broad as to not really be of significant





benefit). Those programmes which have demonstrated success have focused on very specific sub-groups (Yang et al., 2019,
e.g.).

Part of the problem with low success rates for prediction is the presence of prediction bias (Domingos, 2000, e.g.). This can
be attributed to various causes, such as:

- noisy training data

- an incomplete feature set

- strong regularisation

- algorithmic errors

- biased training samples

In order to address these issues, a programme was undertaken to improve the reliability of the machine learning results (Kvam-
men et al.). In addition to using more up-to-date machine learning methods, attention was paid to the provision of a compre-
hensive data set for training the algorithms. As a part of this process, it was deemed important to remove sources of bias in
the classification of the training data set. Following some preliminary work with small samples of both grey-scale and colour
images, a main classification run was undertaken.

As the differences between the classifications of informed researchers was significant, the findings are presented here. It is
intended that they will serve as a reference point for other endeavours in the development of machine learning training sets
for both auroral research, but also for any other field where machine learning image recognition is developed from specialised
sets, categorised by subject experts.

## 2 Methodology

Images from the Kiruna all-sky camera (location: 67.84°N, 20.41°E, 425 m above mean sea level, operated by the Swedish
Institute for Space Physics) were used. Approximately 300 000 of these images from nine winter seasons were filtered down
to a set of 13 947 for human classification (by removing cloudy, moon-lit and twilight images). These were then automatically
cropped to the central 128×128 pixels and data were binned to remove point-like noise (e.g. stars or defective pixels). Two
auroral physicists each classified these 13 947 processed images by hand using different software implementations (one using
Python, the other using MATLAB).

The classification was done according to 10 possible classes, listed in Table 1. These classes were the result of several
iterations of planning, where the two experts, together with a machine learning researcher, identified categories which would
be scientifically useful, possible to discern with a reasonable algorithmic network, and suitable for the sample size available.
An example auroral image (after pre-processing) is shown in Figure 1.

After comparison of results, it was found that the experts only agreed on 54% of the images, with most disagreement being
on which images were suitable for training and which had aurora with a unknown-complicated form. Where it was agreed that





**Table 1.** A set of aurora labels used for the classification.

| Class | Description |
|-------|-------------|
| 0 | Auroral breakup |
| 1 | Coloured aurora |
| 2 | Curled aurora |
| 3 | Arcs or bands |
| 4 | Discrete or irregular |
| 5 | Patchy or stripey |
| 6 | Clipped by image edge |
| 7 | Clear sky / no aurora |
| 8 | Unknown or ambiguous |
| 9 | Rejected (artefacts, problems) |

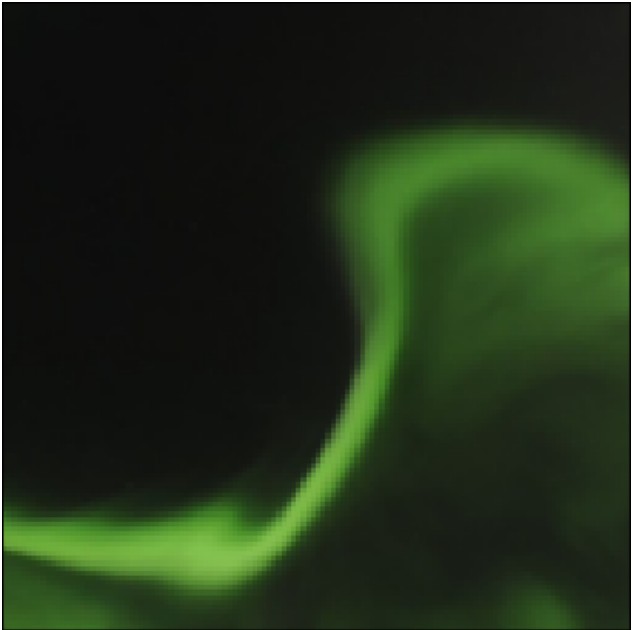

**Figure 1.** Example auroral image after pre-processing (Class 2 = curled).

the image was suitable for algorithm training, the experts agreed on 95% of the labels. By only using the images with agreeing
auroral labels and by excluding the images with ambiguous auroral forms, unwanted features and disagreeing labels, a clean
training data set was produced at the price of excluding approximately 73% of the 13 947 images in the initial data set.



**Table 2.** A set of biases that may affect user classification.

| |
| --- |
| Physical comfort bias |
| Data contrast bias |
| Environment contrast bias |
| Repetition bias |
| Learning bias |
| Feature bias |
| Ambiguity bias |
| Expert knowledge bias |

**Figure 2.** The original right-hand key set for classification. The grey-shaded keys were used for the classification.

## 3   Ergonomic categories

The comparison of the classifications for both the trials and the main classification run allowed the identification of emerging biases based on the approach each researcher took to identify the aurorae in the images. These biases are a result of the levels of comfort (physical and cognitive) that exist during the classification process, leading to the term: "classification ergonomics". Those identified as part of this study are shown in Table 2 and are discussed in the subsequent sections.

### 3.1   Physical comfort bias

The classification of the aurorae in the main study was a 10-class system. Given the designations, the number keys were the obvious choice and the classification software used these, either on the main keyboard (0–9) or the numeric keypad (KP0–KP9). In case of a mistake, it was possible to go back to the previous image, and the backspace key was used to accomplish this. This key configuration is shown in Figure 2.

The first bias that was noted was the inconvenience of the backspace for making corrections. This required moving the right hand completely away from the rest position where the fingers are hovering over the KP4, KP5 and KP6 keys on the keypad.







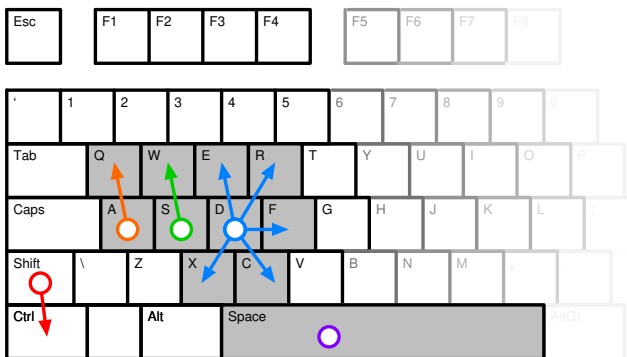

**Figure 3.** A left-hand key set for classification. The coloured circles show the at-rest position of the fingers, with the arrows showing easy-reach positions. The grey-shaded keys were used for the classification. See also Figure 4.

As this was awkward, there was a perceptible reluctance to make corrections. Thus the KP-DECIMAL (to the right of the KP0 key) was used as an alias.

After several hundreds of classifications discomfort was experienced, even with the keyboard rotated 10–20 degrees anti-clockwise to make the keys suit the angle of the right hand. As a result, some testing was also done with more comfortable key arrangements. This resulted in a basic WASD configuration being used. WASD refers to the directional (move forward/backward/left/right) keys as used in FPS (first-person shooter) computer games.

This configuration is shown in Figure 3, where the coloured circles show the at-rest position of the fingers (keys A, S and D), with the arrows showing easy-reach positions. The left thumb rests on the spacebar. The little finger typically can reach the shift and control keys (as a modifier; in FPS games this might be, e.g. run and crouch), but were not used here. The actual keys that were used for the classification are shaded in grey.

Additionally, the keyboard was rotated 10–15 degrees clockwise to match the natural angle of the left wrist and hand, as shown in Figure 4. This was used for most of the classification work and no discomfort was experienced.

### 3.2 Data contrast bias

If the classifier has just seen a faint, patchy aurora, then a following faint, patchy aurora is likely to be classified the same. If the preceding image was a bright break-up, then it is more likely for the faint, patchy aurora to be classified as blank. In the initial parts of the study, attempts were made to mitigate this by normalising the image scale of all images. This was not readily achieved with colour images and thus not pursued. Instead the classifications were done randomly and chronologically, respectively, by the two experts.

### 3.3 Environment contrast bias

Humans naturally retain perceptual constancy. This allows visual features to be discerned against a noisy or changing background: a trait that is useful to all animals in a hunter-prey scenario, for instance. However, this human trait of retaining





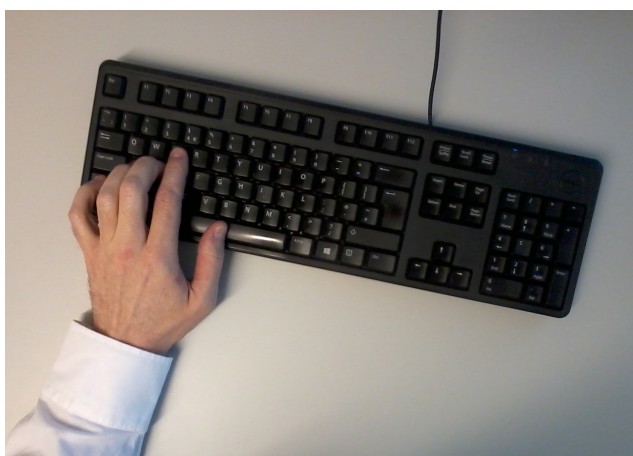

**Figure 4.** Hand position for the left-key set, with the keyboard at an angle of 10–15 degrees, to minimise finger reach strain.

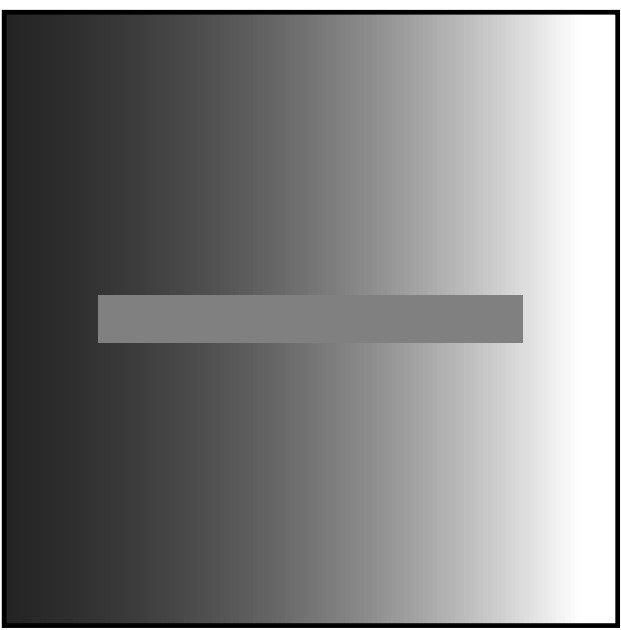

**Figure 5.** Example of environment contrast. The central bar may seem to be a gradient, but it is uniform in shade. Cover the surrounding (real) gradient with paper of uniform colour to demonstrate this.)

perceptual constancy results in optical illusions. Colour constancy and brightness constancy will cause an illusion of colour or contrast difference when the luminosity or colour of the area surrounding an object is changed. The eye partly does this as a result of compensating for the overall lighting (change in the iris aperture), but the brain also compensates for subtle changes within the field of view. An example of this is shown in Figure 5.

Originally, the software presented the image with a white border (the default for the plotting software). However, the contrast made it difficult to discern the difference between features which were faint, but still recognisable, and those which were sub-threshold for visual identification. Hence, the figure background was changed to black. This made it easier to discern the borderline cases.

The environmental conditions beyond the computer screen were also significant. With differences in the ambient lighting

and room brightness being an issue. This was noted and consistency of arrangements sought for the process.

### 3.4 Repetition bias

It is more comfortable to press the same button twice than to press two different buttons. Additionally, if a mistake is made, it is extra effort to go back and correct it. This "laziness" accumulates during the classification process, making long sessions problematic.

For example, if there are 10 similar images in a row, the chance of classifying number 11 in the same way is higher, than if there were 10 random images first. In the study single-repetition bias was 27%, rising to 40% for double-repetition bias.

### 3.5 Learning bias

If there are lots of categories, the classifier may not necessarily hold all of them in mind. Thus some "sectors" of the classification may have a higher activation energy than others. For example, classifying 100s of arcs and patchy aurorae and then

getting a curled case; the classifier may subconsciously think: its not patchy, so it must be an arc... inadvertently omitting the thought of a different class. This is a recency effect (where a new classification is biased toward the set of most recently used labels) which has been reported in the biological sciences (Culverhouse, 2007).

### 3.6 Feature bias

The classifier is more likely to get the classification of a prominent feature correct, than faint or diffuse features. This leads to

a form of confusion bias; e.g. what to do with a bright curl aurora (Class 2) on a background of diffuse patchy aurora (Class 5).

There is also positivity bias, where identification is biased by prior expectations (Culverhouse, 2007). In auroral classification, the substorm progression (development of the auroral display) makes it possible to anticipate the next image. This is partially mitigated by randomising the samples, but this can lead to contrast bias (Section 3.2).

### 3.7 Ambiguity bias

Ambiguity bias occurs when there is a confusion as to what a particular image may be. This is exacerbated by feature bias (Section 3.6). However, even in cases where there is no dominant feature, the classifier will tend to subconsciously identify some feature and latch onto that, to the exclusion of other features in the image. For the main study, it was decided that ambiguous images should be rejected, to make the learning environment clearer for the machine learning algorithms — thus


such ambiguities were undesirable. What was noted was that, especially early in the study, the users would tend to try to classify the auroral image rather than reject it. When it was clear that there was no shortage of data, this tendency reduced.

Nevertheless it is recommended that there is a clarification of classification rules, making it clear what the user should do in a case of mixed features. If there is a precedence or priority of forms, then that should also be made very clear. Even so, there will remain ambiguities and borderline cases. When data volume allows, these could be discarded.

### 3.8 Expert knowledge bias

Differences in "expert knowledge" that affect the results have also been seen. For example, although the two expert researchers involved are knowledgeable in auroral physics and its optical manifestation in general, one had done research on auroral arcs, whereas the other has not. The specialist was more picky on the arc classification (classifying 15% fewer), partially as a result of having a deeper understanding of the underlying physics, but partly in terms of having seen many more images prior to approaching the classification task. This led to a higher level of discernment on that particular category.

## 4   Discussion

The application of machine learning to auroral classification is an area in which only a few studies have been carried out. However, development is now progressing rapidly and it is likely that it will be applied much more and become an important part of auroral research in the future. Therefore, it is vitally important to properly address the ergonomics and biases sooner rather than later, in order to avoid inadvertently introducing errors and biases early in the establishment of this new area of science.

The discrepancy between expert classifiers has been reported before. A previous auroral study had two experts that agreed on the class in about 70% of the images and that the experts chose the unknown class in almost 50% of all images (Syrjäsuo et al., 2007). However, analysis of potential reasons for the discrepancy were not included. Similarly, a biological study, found that trained personnel achieve 67% to 83% self-consistency and 43% consensus between people in expert taxonomic labelling tasks, with those routinely engaged in particular discriminations returning accuracies in the range of 84% to 95% (Culverhouse et al., 2003).

It is surmised that, in addition to ambiguity over the content, there is an ergonomic factor that contributes to classification bias. In any general image classifications (e.g. car vs house, or tree vs dog), common knowledge, massive samples of people doing the training, and clear-cut distinctions between the objects, makes it easier (although not completely) to avoid subjective bias, or even prejudice. But when the classification is being done by a small number of experts, with built-in knowledge and subject background, then the training set can readily become subject to inadvertent bias. However, as a specialist field, there may be no choice. The general public may not be able to know the difference between auroral types (at least not without some training... itself subject to interpretation).

Four key human traits that affect classification performance are: (a) a short-term memory limit of 5–9 items, (b) boredom and fatigue, (c) recency effects where a new classification is biased toward the set of most recently used labels, and (d) positivity



bias, where identification is biased by prior expectations (Culverhouse, 2007). Ambient noise, high ambient temperature, difficulty of discerning auroral features, and lack of sleep decrease performance. Additionally, attention should be paid to error analysis and associated quality metrics to weight not just algorithms, but also human-based classification, according to performance (Zhu et al., 2014).

**5   Conclusions**

Ergonomics refers to the design factors intended to improve productivity by reducing the fatigue and discomfort of the user. As part of the ongoing study, the trade off between user fatigue and scientific bias is considered. When considering the training of a classification scheme, it is important to reconcile the aspects of the task which cause scientific bias, but which improve overall efficiency. Given the nature of large classification programmes, removing sources of repetitive and cognitive strain not 165 only serve to improve the working condition of the user, but also assist in ensuring that no work environment bias is injected into training data sets that are later used for classification.

This is of particular use for specialist fields (such as auroral research), where it is necessary to use a small number of experts to train algorithms. Consensus of opinion on any given classification is important in reducing errors in the training set, yet it is typical for experts to operate in very small teams, or even alone. Addressing these issues will help future studies drive a balance 170 between the statistical effectiveness of large samples and the potential for scientific bias which may result from inappropriate ergonomic design that facilitates large sample classifications. This is particularly important for auroral research, where the application of machine learning is relatively new, and there is much potential for misguided research on the grounds of biased input data.

*Data availability.*   Original source images are publicly available from the Swedish Institute for Space Physics archive 175 `http://www2.irf.se/allsky/`. The classification from these images by the authors is available from the University of Tromsø Space-physics Experiment, Information and Data website `http://seid.uit.no/data/` and has been submitted as supplementary material with this manuscript.

*Author contributions.*   DM and AK jointly established the classification definitions. DM and AK independently developed their classification tools and applied them to the data set. The text and figures for this manuscript were produced by DM, with comments and editing by AK.

*Competing interests.*   The authors declare that they have no conflict of interest.





*Acknowledgements.* The authors wish to thank K. Wickstrøm, N. Partamies, C. Negri, T. Paavilainen and A. Panther for their contributions to this work. In addition, the authors would like to thank U. Brändström and the Swedish Institute of Space Physics for providing the original auroral image data. A. Kvammen is supported by the Tromsø Research Foundation.



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
