# Peer review of "Auroral Classification Ergonomics and the Implications for Machine Learning"

_Geoscientific Instrumentation, Methods and Data Systems, 2019_

## Referee Comment (RC1) · Anonymous Referee #1 · 16 Mar 2020

This paper addresses the sources of expert-injected bias in the training samples for auroral classification. By developing an auroral training data set, the authors listed the bias to serve as a checklist for improving the training data integrity. It is an interesting point for machine learning researchers. However, the organization of the manuscript needs improvement to get published. My concerns and questions are listed below.

Major issues: This paper reads like neither a behavior study paper nor a data paper because it does not provide much quantitative results about the annotation procedure or the resultant dataset. Providing a checklist of the expert-related biases does not make much contribution since these biases have been reported in psychology and cognition research. I would like to see a data paper like that about ImageNet, PASCAL VOC in the machine learning field.

Minor issues: 1. Section 2, the original images were "cropped to the central 128x128 pixels and data were binned ...", was the image cropped to 128x128 first and then binned to 64x64 or less, or the cropped image was smoothed by binning? The description is not clear. In addition, what is the criteria for choosing the parameter 128? 2. Section 2, two physicists processed the images using different software implementations, did they follow the same protocol? Why did not they use a same software to exclude unnecessary discrepancy? 3. Section 2, a brief introduction to the characteristics of each category is expected. Showing example images of each category is helpful to more intuitively explain the problems encountered during classification, such as data contrast bias and environment contrast bias. 4. Section 2, images not suitable for training machine learning algorithm were also removed. What is the standard of suitable for training? 5. Section 3.2, the classifications were done randomly and chronologically, respectively, by the two experts. However, classify chronologically seems more likely to encounter the situation that authors mentioned 'If the classifier has just seen a faint, patchy aurora, then a following faint, patchy aurora is likely to be classified the same'. 6. It is expected to see some experimental results on the labelled dataset using currently available machine learning methods, which can be treated as a baseline for further research. 7. Since the authors provided the data list in the supplement, I tried to download the original images accordingly. Unfortunately, 4767 images in the 13947 list are not exist on the website (http://www2.irf.se/allsky/). The web link provided in the manuscript (http://seid.uit.no/data/) is broken. As a researcher in auroral image analysis, I do hope the authors publicly release the dataset completely (original images + annotations).

---

## Referee Comment (RC2) · Anonymous Referee #2 · 2 Apr 2020

**GENERAL COMMENT**

The present manuscript addresses an important scientific and technical question, i.e. the one related to the potential source of prejudice in machine learning with particular reference to the bias in the training samples. The work is based on a dataset of auroral images captured in Sweden. The topic fits the scope of the journal and is interesting for the readers. Although the study seems to be relevant and displays some promising results, the lack of rigour and essential details are problematic. My opinion is that the paper needs a big improvement before being reconsidered for discussion and eventually, publication. In the following, I list some specific comments that should be addressed carefully.

**SPECIFIC COMMENTS**

Section 1.

1) Although the application of machine learning approaches is relatively new for "aurora" purposes, a brief review of cognitive and physical bias in previous studies, also in other fields, is expected. 2) Aurora phenomenon is not described, as well as the different morphological types. I think that for an inexpert reader can be difficult to go in the core of the topic. I suggest to introduce a brief explanation of the different morphological categories (i.e., in table 1). Also, showing more examples of auroral images (i.e., in Section 2), at different times, can be really appreciated.

Section 2.

3) Information is needed about how the pictures were taken and their temporal interval. 4) Details are missing about all the pre-processing phase. Just an example of the final pre-processed image is shown, but no information is provided about all stages of image preparation. I think this part is crucial, having a role in the classification bias. 5) It is essential to clarify how the categories are defined. Is there some overlap between the defined classes? For the trained human eye, there is a clear distinction between the different classes? Authors refer to ambiguous forms and unexpected features, what are they? I suggest introducing some examples. 6) It is not clear why the auroral physicists classify the images using different software. Please motivate this choice.

Section 3.

7) The two experts classify the same dataset in two different ways: randomly and chronologically. The quantitative analysis of the results, obtained in the two way separately, is expected concerning their influence in some category of bias. 8) The authors list and describe different sources of potential bias observed during the classification. However, this analysis is qualitative, and it is not clear the impact of each category respect the results. 9) An important aspect related to the study of cognitive biases is the validation of strategies for mitigation of their effects in cases when they lead to incorrect judgment. In the manuscript, authors provide strategies for mitigating the negative

effects of bias but, unfortunately, not for all the categories investigated.

---

## Author Comment (AC1) · 7 May 2020

This document pertains to the paper *Auroral Classification Ergonomics and the Implications for Machine Learning* and the comments made by the reviwers. The authors would like to thank the reviewers for their constructive criticism. We appreciate the time they have taken to help us improve this work. Included below are the reviewers' comments (shown in *green italics*) with the authors' responses interleaved.

— *D. M^cKay & A. Kvammen, 07-May-2020*

[Figure]

**Response to Reviewer 1**

**Reviewer overview:** "*This paper addresses the sources of expert-injected bias in the training samples for auroral classification. By developing an auroral training data set, the authors listed the bias to serve as a checklist for improving the training data integrity. It is an interesting point for machine learning researchers. However, the organization of the manuscript needs improvement to get published. My concerns and questions are listed below.*"

—

**Reviewer major issues:** "*This paper reads like neither a behavior study paper nor a data paper because it does not provide much quantitative results about the annotation procedure or the resultant dataset. Providing a checklist of the expert-related biases does not make much contribution since these biases have been reported in psychology and cognition research. I would like to see a data paper like that about ImageNet, PASCALVOC in the machine learning field.*"

We fully agree that a data analysis paper is essential in this field of research and that such a study should be done. However, we believe that this project and the $\approx$14 000 image data set is way to small too conduct such a study. Both ImageNet and PAS-CALVOC are large, multi-institutional projects that extend over several years with data sets containing hundreds of thousands images. The motivation for this paper is not to analyse the data set itself, but to report our experience after labelling $\approx$14 000 auroral images.

As the reviewer rightly notes, the reported biases have been suggested in previous psychology and cognition research. However, it is doubtful that auroral researchers that intend to do similar classification projects will seek and study such papers. We

therefore think that, in addition to our findings, "providing a checklist of the expert-related biases", specifically relevant for aurora classification, is a useful contribution in this growing field of research.

**Reviewer comment 1.1:** "*Section 2, the original images were "cropped to the central 128x128pixels and data were binned...", was the image cropped to 128x128 first and then binned to 64x64 or less, or the cropped image was smoothed by binning? The description is not clear. In addition, what is the criteria for choosing the parameter 128?*"

Yes, agreed. This section was poor and has been re-written to include these details.

**Reviewer comment 1.2:** "*Section 2, two physicists processed the images using different software implementations, did they follow the same protocol? Why did not they use a same software to exclude unnecessary discrepancy?*"

As the parent study (Kvammen 2020, referred to in the paper) was exploring machine learning, we were keen to use training data that had some level of robustness. The problem with this volume of classification was finding time and expertise to carry it out. The authors had thus deliberately chosen to *both* classify all the data, and then compare the results... using only those images where we both independently agreed. So, the protocol was the same, but then we deliberately pursued implementation of that protocol independently to cross-check the results.

That, in turn, also led to a difference in implementation language (MATLAB v Python) due to each of us having different experience with programming. The algorithms themselves were not complicated; they were merely a way of presenting the data and accumulating the classification. The reviewer is correct that the same software could have
been used, but we were independently checking each other's results as we proceeded with the work. A comment to this effect has been added to the paper.

**Reviewer comment 1.3:** "*Section 2, a brief introduction to the characteristics of each category is expected. Showing example images of each category is helpful to more intuitively explain the problems encountered during classification, such as data contrast bias and environment contrast bias.*"

A brief description of each label is added to Table 1 and sample images of each label is now presented in Figure 1 in the revised manuscript.

**Reviewer comment 1.4:** "*Section 2, images not suitable for training machine learning algorithm were also removed. What is the standard of suitable for training?*"

We clarify this in the table with the set of aurora labels used for the classification run. We were rejecting cases as unsuitable when a) the case was complex or ambiguous, b) it was obviously messed up (bad light pollution, rain drops, snow, or cloud).

Thereafter we were looking for cases where the experts agreed. This has now been clarified in the text.

**Reviewer comment 1.5:** "*Section 3.2, the classifications were done randomly and chronologically, respectively, by the two experts. However, classify chronologically seems more likely to encounter the situation that authors mentioned 'If the classifier has just seen a faint, patchy aurora, then a following faint, patchy aurora is likely to be classified the same'.*"

Agreed. Additionally, it can work both ways. Seeing high-contrast after low-contrast may adversely affect the result as seeing low-contrast after low-contrast. This is compounded by the intrinsic knowledge that the classifier may have of the phenomenon.

Classifying randomly, and multiple times, would establish one data set. This could be compared to chronologically and, importantly, reverse-chronologically sorted images. This repeated classification is way beyond the time limits of the project, but we have mentioned this in the text to ensure that the idea is there for future consideration.

**Reviewer comment 1.6:** "*It is expected to see some experimental results on the labelled dataset using currently available machine learning methods, which can be treated as a baseline for further research.*"

The experimental results of the labelled dataset are presented in Kvammen 2020. The data set has been published separately https://dataverse.no/dataset.xhtml? persistentId=doi:10.18710/SSA38J (see also our response to the next point). This has now been added to the text.

**Reviewer comment 1.7:** "*Since the authors provided the data list in the supplement, I tried to download the original images accordingly. Unfortunately, 4767 images in the 13947 list are not exist on the website (http://www2.irf.se/allsky/). The web link provided in the manuscript (http://seid.uit.no/data/) is broken. As a researcher in auroral image analysis, I do hope the authors publicly release the dataset completely (original images +annotations).*"

We are puzzled by the report that 4767 images in the 13947 list did not exist. We definitely obtained the images at the time of the project and have checked again as we

have prepared this response letter. The validation script (Bash, tested on Ubuntu Linux LTS18.04) that we used to do this is included in this response as an appendix. We ran the script on the morning of 06-May-2020 and verified that all 13947 images that we listed in our supplementary material to the paper were present.

The http://seid.uit.no/data/ link was temporary and it unfortunately expired before the review process was complete. We have since established a public permanent link to the data at a properly curated site: https://dataverse.no/dataset.xhtml?persistentId= doi:10.18710/SSA38J . This link has been added to the data sources section.

**Response to Reviewer 2**

**Reviewer overview:** "*The present manuscript addresses an important scientific and technical question, i.e. the one related to the potential source of prejudice in machine learning with particular reference to the bias in the training samples. The work is based on a dataset of auroral images captured in Sweden. The topic fits the scope of the journal and is interesting for the readers.*"

—

**Reviewer major issues:** "*Although the study seems to be relevant and displays some promising results, the lack of rigour and essential details are problematic. My opinion is that the paper needs a big improvement before being reconsidered for discussion*

*and eventually, publication. In the following, I list some specific comments that should be addressed carefully.* "

This has been noted, and we address the issues in the following responses. Additionally, our responses to Reviewer #1 address some of the issues raised.

**Reviewer comment 2.1:** "*Section 1 / Although the application of machine learning approaches is relatively new for "aurora" purposes, a brief review of cognitive and physical bias in previous studies, also in other fields, is expected.* "

We include references to some other field papers in the Discussion section. We have now also mentioned this in the introduction to alert the reader in advance that this material is covered in the paper.

**Reviewer comment 2.2:** "*Aurora phenomenon is not described, as well as the different morphological types. I think that for an inexpert reader can be difficult to go in the core of the topic. I suggest to introduce a brief explanation of the different morphological categories (i.e., in table 1). Also, showing more examples of auroral images (i.e., in Section 2), at different times, can be really appreciated.* "

We agree and these issues have been addressed in the revised manuscript. A description of the auroral phenomenon has been included in the introduction. A brief explanation of the labels is added to Table 1 and sample images are presented in Figure 1.

**Reviewer comment 2.3:** "*Section 2/ Information is needed about how the pictures*

*were taken and their temporal interval.* "

Images are a 6 second exposure, taken every 60 seconds (at the start of each minute). This has been added to the text, along with details of the camera used, as well as a reference.

**Reviewer comment 2.4:** "*Details are missing about all the pre-processing phase. Just an example of the final pre-processed image is shown, but no information is provided about all stages of image preparation. I think this part is crucial, having a role in the classification bias.* "

Agreed. This section was sloppy and has been re-written to include these details.

**Reviewer comment 2.5:** "*It is essential to clarify how the categories are defined. Is there some overlap between the defined classes? For the trained human eye, there is a clear distinction between the different classes? Authors refer to ambiguous forms and unexpected features, what are they? I suggest introducing some examples.* "

A brief description of each category is included in Table 1. Clearly overlapping cases are labelled as "unknown-complicated" and were not used for training the neural networks. This is where there are two distinct features which would otherwise be classified separately — thus, ambiguous. The "unexpected forms" refers to features that are not expected in a clean training data set (such as light pollution or obstructing objects).

Sample images of "unknown-complicated" aurora (ambiguous forms) and "rejected" (unexpected features) are added to Figure 1 to make this clear. However, we have deliberately kept this brief as the point of the paper is not to examine the auroral features, but the biases that occur during the labelling of images.

**Reviewer comment 2.6:** "*It is not clear why the auroral physicists classify the images using different software. Please motivate this choice.* "

We have provided explanation of this now in the Methodology section. See also our response to Reviewer comment 1.2.

**Reviewer comment 2.7:** "*Section 3 / The two experts classify the same dataset in two different ways: randomly and chronologically. The quantitative analysis of the results, obtained in the two way separately, is expected concerning their influence in some category of bias.* "

Yes, this has been addressed by providing additional text to the "Data Contrast Bias" section of the paper.

**Reviewer comment 2.8:** "*The authors list and describe different sources of potential bias observed during the classification. However, this analysis is qualitative, and it is not clear the impact of each category respect the results.* "

To isolate each case and look for the effect would be a large study. This is not tractable for this project. The biases were identified, based on our observations during the labelling run. In addition, the impact of each category is not general, but will will vary from person to person and depend on the considered classes and images at hand. Thus, providing quantitative results of the bias impact will not contribute much to the paper.

**Reviewer comment 2.9:** "*An important aspect related to the study of cognitive biases is the validation of strategies for mitigation of their effects in cases when they lead to incorrect judgment. In the manuscript, authors provide strategies for mitigating the negative effects of bias but, unfortunately, not for all the categories investigated.* "

We have addressed this and offer recommendations for all cases.

**Additional changes**

Since the initial submission of the paper the classification system was revised. This resulted in 9 categories, rather than 10. This has no bearing on the findings presented in this paper — this is because we are looking are the causes of bias in labelling images, rather than the nuances of the divisions between those classes. However, we have updated the keyboard figures and the category tables to reflect what was actually used in the parent study. As mentioned, this does not impact our conclusions; the biases and sources of error remain the same.

* * *

**Appendix: File validation script**

```bash
**!/bin/bash**
**Check that the files in the submitted supplementary material are available.**
**Sample URL =**
**http://www2.irf.se/allsky/2019/20190102/jpgs/KRN20190102T154500E06000Q.JPG**

LIST=/tmp/tmp_aurnn.txt
cat aurnn_class_20191120a.csv | grep ^KRN20 |\
    sed 's/,/ /g' | awk '{print $1}' > $LIST

**Set up some loop counters**
COUNT=0
NOTFOUND=0
START='date'

**Loop over all the files in the list**
for FILE in 'cat $LIST'
do
  # Print a progress indicator
  if [ $((COUNT%50)) == 0 ]
  then
    printf "\n%5d : " $COUNT
  fi

  # Determine the URL components
  YEAR='echo $FILE | awk '{print substr($1,4,4)}''
  DATE='echo $FILE | awk '{print substr($1,4,8)}''
  HOUR='echo $FILE | awk '{print substr($1,13,2)}''

  # Directories for morning hours are actually the previous day
  if [ $HOUR -lt 12 ]
  then
    T1='date -d $DATE +"%s"'
    T2=$((T1-86400))
    DATE='date -d @$T2 +"%Y%m%d"'
```

```
  fi

  # Build the URL
  URL=http://www2.irf.se/allsky/$YEAR/$DATE/jpgs/$FILE

  # Use a HEAD request to verify that the URL exists
  wget --spider -q ${URL}

  # Look at the return code and report if there was an error
  if [ $? != 0 ]
  then
    echo ""
    echo $URL
    let NOTFOUND=NOTFOUND+1
  fi
  echo -n "."
  let COUNT=COUNT+1
done

**Make the final report was what was not found**
echo
echo "Script complete : $NOTFOUND files were not found"
echo "      Started at : $START"
echo -n "   Completed at : "
date
**EOF**
```

---

## Author Response (AR2)

This response document pertains to the paper *Auroral Classification Ergonomics and the Implications for Machine Learning*. The authors would like to thank the reviewers and editor for taking the time to re-review the work. Below is the response to the requests made.

— *D. McKay & A. Kvammen, 06-Jun-2020*

**Response to Editor**

**Editor request:** "*Based on the reviewers' comments, the article was considerably improved and is now acceptable for publication. I agree with one reviewer that the title should be edited to: "Technical Note: Auroral Classification Ergonomics and the Implications for Machine Learning".*"

We agree to the requested change.

**Response to Reviewer 1**

**Reviewer comment:** "*For final publication, the manuscript should be accepted as is.*"

No response required.

**Response to Reviewer 2**

**Reviewer comment:** "*many details are included and the paper have certainly improved from the first version. However, I have still some doubts about it, since my main comment is not properly assessed: it is not clear to what extent the paper is presenting some new findings supported by quantitative information. The manuscript is still dominated by the presentation of a checklist of biases for auroral classification and some strategies for mitigation of their effects but it does not provide much quantitative results. Therefore, it is difficult to evaluate the scientific soundness of the work. I agree that this contribution should have practical value for auroral research but can be not considered totally a novelty because have been suggested in previous psychology and cognition studies. Thus, at the current form, I would suggest to present this work as "Technical note" rather than as "Research paper".*"

We acknowledge the suggestion made to present the paper as a Technical Note. The title has been changed accordingly to reflect this, in accordance with the Editor's request, above.